

# Function of JARID2 in bovines during early embryonic development

Yao Fu[*], Jia-Jun Xu[*], Xu-Lei Sun, Hao Jiang, Dong-Xu Han, Chang Liu, Yan Gao, Bao Yuan and Jia-Bao Zhang

Department of Laboratory Animals, College of Animal Sciences, Jilin University, Changchun, Jilin, China
[*] These authors contributed equally to this work.

## ABSTRACT

Histone lysine modifications are important epigenetic modifications in early embryonic development. JARID2, which is a member of the jumonji demethylase protein family, is a regulator of early embryonic development and can regulate mouse development and embryonic stem cell (ESC) differentiation by modifying histone lysines. JARID2 can affect early embryonic development by regulating the methylation level of H3K27me3, which is closely related to normal early embryonic development. To investigate the expression pattern of JARID2 and the effect of JARID2-induced H3K27 methylation in bovine oocytes and early embryonic stages, JARID2 mRNA expression and localization were detected in bovine oocytes and early embryos via qRT-PCR and immunofluorescence in the present study. The results showed that JARID2 is highly expressed in the germinal vesicle (GV), MII, 2-cell, 4-cell, 8-cell, 16-cell and blastocyst stages, but the relative expression level of JARID2 in bovine GV oocytes is significantly lower than that at other oocyte/embryonic stages ($p < 0.05$), and JARID2 is expressed primarily in the nucleus. We next detected the mRNA expression levels of embryonic development-related genes (OCT4, SOX2 and c-myc) after JARID2 knockdown through JARID2-2830-siRNA microinjection to investigate the molecularpathwayunderlying the regulation of H3K27me3 by JARID2 during early embryonic development. The results showed that the relative expression levels of these genes in 2-cell embryos weresignificantly higher than those in the blastocyst stage, and expression levels were significantly increased after JARID2 knockdown. In summary, the present study identified the expression pattern of JARID2 in bovine oocytes and at each early embryonic stage, and the results suggest that JARID2 plays a key role in early embryonic development by regulating the expression of OCT4, SOX2 and c-myc via modification of H3K27me3 expression. This work provides new data for improvements in the efficiency of *in vitro* embryo culture as well as a theoretical basis for further studying the regulatory mechanisms involved in early embryonic development.

# INTRODUCTION

Epigenetic modifications are common biological phenomena involved in gene regulation and are essential for maintaining the normal physiological activities of mammals. Epigenetic modification refers to the heritable modification of gene expression, without any changes

Corresponding authors
Bao Yuan, yuan_bao@jlu.edu.cn
Jia-Bao Zhang, zjb515@126.com

in the DNA sequence. Many studies have shown that changes in a variety of epigenetic modifications occur during mammalian embryonic development. Early embryonic development is the period in which cell division and differentiation are most active, and there is large-scale transcription of zygotic genes. At this stage, genes are subject to activation and inhibition under the control of signals from both inside and outside of cells, and many regulatory mechanisms participate in their regulation. Studies conducted in recent years have increasingly shown that epigenetic regulation plays an important role during mammalian embryonic development.

Epigenetic modifications mainly include DNA methylation and histone modification, which generally have a synergistic effect on mammalian gene expression regulation (*Delcuve, Rastegar & Davie, 2009*). Acetylation and methylation are the most important factors in histone modification. DNA is methylated via the transfer of a methyl group (CH3) from S-adenosylmethionine (SAM) to cytosine by DNA methyltransferase (*Jurkowska, Jurkowski & Jeltsch, 2011*). This covalent modification may change gene expression by altering chromatin condensation and heterochromatin formation and generally inhibits gene expression (*Brenet et al., 2011*; *Choy et al., 2010*). DNA methylation is critical for the development and reprogramming of mammalian embryos and mainly plays roles in regulating gene expression, genomic imprinting, transposon inactivation, X chromosome inactivation and alternative splicing of mRNA (*Cotton et al., 2011*; *Dean, Lucifero & Santos, 2005*; *Isagawa et al., 2011*; *Malousi, Maglaveras & Kouidou, 2008*; *Verona, Mann & Bartolomei, 2003*; *Walsh, Chaillet & Bestor, 1998*). In general, DNA methylation is negatively correlated with gene expression, and there is a negative correlation between the methylation of a gene and its expression. Histones are an important component of eukaryotic nuclear chromosomes. There are many amino acids in histones that can be covalently modified by reactions including methylation, acetylation, and ubiquitination. These amino acids may cause some local chromatin structures to "open" or close"; they can also determine the specific area of DNA to which special proteins can bind. Such modifications can promote normal transcription, reproduction and repair. Acetylation and methylation of histones are the most common and important epigenetic modifications. Histone methylation modifications promote or inhibit gene expression, depending on the site as well as the degree of methylation. Different methylation sites and different degrees of methylation present different functions. In summary, histone methylation-modifying enzymes play an important role in early embryonic development. Histone methylation and demethylation exist in a state of dynamic equilibrium. These processes are catalyzed by histone methyltransferase (HMT) and histone demethylase (HDMT), respectively. Modifications performed by histone methyltransferase can cause the nucleosome to directly adjust the structure of chromatin, thus regulating the expression of related genes.

Histone modification plays an important role in early embryonic development. For example, in mouse oocytes, histones may maintain acetylation until oocyte follicular lysis at the germinal vesicle (GV) stage (*Kim et al., 2003a*). In sheep oocytes, although no acetylation of H3K9 is detected from the GV stage to the MII stage, high levels of acetylation are detected in male and female pronuclei at the zygote stage, and the degree of acetylation is similar to that in mouse zygotes (*Hou et al., 2008*). During the maturation of

mouse oocytes, H3K9 maintains a constant degree of methylation, while methylation does not appear in the male chromosome group until the first mitosis after fertilization (*Yeo et al., 2005*). It was found that the degree of H3K27 methylation affects early embryonic development, and the degree of H3K27me3 methylation is closely related to normal early embryonic development (*Marinho et al., 2017*; *Ross et al., 2008*; *Zhang et al., 2009*).

Lysine demethylases, which include the LSD family and the (jumonji C) JmjC domain family, play a role in early embryonic development. The JmjC family includes the most types of methyltransferases, each of which has a JmjC domain; most of these methyltransferases enable histone lysine demethylation using Fe(II) and α-ketoglutarate (αKG) as co-enzyme factors (*Jackson et al., 2003*). Unlike other families of demethylases, JHDMs can remove all histone lysine methylation moieties. JARID proteins are members of the JmjC methyltransferase family and have been widely studied. The JARID2 (jumonji, AT-rich interactive domain 2) protein is one member of the jumonji demethylase protein family; unlike other members in this family, JARID2 may not exhibit demethylase activity (*Kooistra & Helin, 2012*). JARID2 was first identified in mice as a regulator of neural development (*Jung, Mysliwiec & Lee, 2005b*; *Takeuchi et al., 1995*); it was shown to interact with most inhibitory complex 2 to regulate the self-renewal ability of embryonic stem cells (*Herz & Shilatifard, 2010*; *Pasini et al., 2010*). The absence of JARID2 reduces the transferability of mouse hematopoietic stem cells and embryonic stem cells (ESCs)line 17 (*Kinkel et al., 2015*). At the molecular level, JARID2 interacts with histone methyltransferases to regulate the expression of target genes (*Cai et al., 2013*; *Kim et al., 2003b*; *Margueron et al., 2009*). JARID2 also plays an important role in the normal development of mouse embryos and the differentiation of ESCs.

However, the epigenetic modifications that occur during early bovine embryonic development are still unclear. Thus, it is important to determine whether JARID2 exerts an effect on bovine embryonic development and determine the mechanism of regulation. This study explores the expression pattern of JARID2 at all bovine oocyte and early embryonic stages and its mechanism of action, including its influence on H3K27 methylation.

## MATERIALS AND METHODS

### Ethics statement

The experiments were performed strictly according to the guidelines of the Guide for the Care and Use of Laboratory Animals of Jilin University. In addition, all experimental protocols were approved by the Institutional Animal Care and Use Committee of Jilin University (Permit Number: 20160522).

### Collection of bovine oocytes and *in vitro* maturation

Bovine ovaries were collected from the No. 3 Slaughter House of the Haoyue Group in Changchun City. After collection, they were placed in vacuum bottles containing normal saline at 37 °C and then sent to the laboratory within 3–5 h. After cleaning the ovaries with normal saline containing penicillin and streptomycin (Sigma, St. Louis, MO, USA) at 37 °C, follicular fluid was extracted from the 2–8 mm normal follicles on the surface of the ovaries and then slowly injected into a 50-ml centrifuge tube. The collected follicular fluid was left

undisturbed for 15 min to allow the cells to settle at the bottom of the centrifuge tube. The precipitated cells were subsequently extracted and placed in a plate containing washing solution (10% PVA) (Sigma, St. Louis, MO, USA). The cumulus oocyte complexes (COCs) with more than 3 layers of cumulus cell coats were selected under a microscope, quantified and collected in drops of culture medium for *in vitro* oocyte maturation; 18–20 COCs were placed in each drop and transferred to a 38.5 °C incubator with 5% $CO_2$ for 24 h.

### *In vitro* fertilization

Frozen semen (Simmental semen from the Yanbian Animal Husbandry Development Group Co., LTD) straws were quick-thawed in a 37 °C water bath, and the thawed semen was then transferred to equilibrated Dulbecco's phosphate-buffered saline (DPBS). After the gently shaking the semen, it was centrifuged for 3 min at 1,300 rpm; this process was repeated twice. Then, the semen was resuspended and centrifuged for 3 min at 1,300 rpm. The precipitated semen was transferred to a straw containing equilibrated *in vitro* fertilization solution, which was subsequently placed in an incubator at 38.5 °C with 5% $CO_2$ for 30 min, allowing the sperm to complete the swimming up process. Then, liquid was collected from the upper layers and inspected with a microscope to calculate sperm viability and density. After washing with *in vitro* fertilization solution, the mature COCs and the sperm acquired after swimming up were transferred to equilibrated fertilization drops; each of these droplets contained 15 COCs, and they were placed in an incubator at 38.5 °C with 5% $CO_2$ for fertilization. After 24 h, the cumulus cells were removed using 0.1% hyaluronidase (Sigma, St. Louis, MO, USA) enzyme solution.

### *In vitro* cultivation

Fertilized embryos were selected and washed three times with culture medium and then transferred to *in vitro* cultivation (IVC) drops that were equilibrated in advance for further cultivation. Each drop contained 18–20 fertilized embryos. The embryos were collected at different developmental stages according to the experimental design.

### Cell transfection

Bovine cumulus cells were cultivated in 24-well plates and transfected as follows when cell fusion reached 75%: based on ratio of 2 µl of Lipofectamine[TM] 2000 (Invitrogen, USA) to 1 µl of siRNA (Gene Pharma, Shanghai, China) per well of a plate, 100 µl of serum-free DMEM/F12 medium (Gibco, Waltham, MA, USA) was added to each well. The plate was then incubated at room temperature for 5 min, after which Lipofectamine[TM] 2000 and the siRNA mixture were added, followed by incubation at room temperature for 20 min. Finally, 200 µl of the siRNA–Lipofectamine[TM] 2000 mixture was added to each well, and the cells were incubated at 37 °C. Eight hours later, the cells were transferred to complete medium containing 10% fetal bovine serum (BI, Beit HaEmek, Israel) and then collected after 36 h.

### Microinjection of siRNA for JARID2 knockdown

Twenty-four hours after *in vitro* fertilization, the COCs were treated with 0.1% hyaluronidase to remove the surrounding cumulus cells. Then, the COCs were washed

three times with fresh solution to ensure that all cumulus cells were removed. The zygotes were next randomly divided into two groups and transferred to IVC drops containing CB buffer; the concentration of siRNA was maintained at 10 $\mu$M. After installing the syringe needle and holding pipette, the operating wall was adjusted to start the injection. After microinjection, the zygotes were transferred to IVC drops that were equilibrated in advance; each drop contained 15 zygotes, which were collected at different stages.

## Immunofluorescence detection

Immunofluorescence staining of embryos was performed as follows. Early embryos at different stages were placed in 24-well Petri dishes containing 4% paraformaldehyde and fixed at room temperature with ventilation for 30 min. After fixing, the embryos were permeabilized with 0.5% Triton X-100 at 4 °C for 20 min. Then, the early embryos were transferred to a 1% BSA solution and incubated at 37 °C for 1 h. The primary antibody (Abcam, Cambridge, UK) was diluted 200 times with a 1% BSA solution, followed by incubation with the samples overnight at 4 °C. Secondary antibodies (Boston, MA, USA) were also diluted with 1% BSA solution, followed by incubation in the dark with the samples for 1 h in the incubator at 37 °C. Next, the samples were incubated with the DAPI (Sigma, St. Louis, MO, USA) fluorescence staining reagent, diluted 1,000 times, 2 min at room temperature. Then, the slides were sealed, and the expression patterns of the target proteins were observed. The fluorescence staining intensity of early embryos at different developmental stages was detected with fluorescence intensity analysis software.

In the abovementioned immunofluorescence staining process, the samples were repeatedly washed at each step.

## Immunofluorescence intensity analysis

Photographs of each experimental group were acquired using the same parameters. The mean fluorescence intensity in the nuclear region was used as the measurement index for changes in H3K27me3. We performed statistical analyses of the mean fluorescence intensity in the early embryonic nuclei at the 2-cell stage, 4-cell stage, 8-cell stage and blastocyst stage in the JARID2 siRNA silencing group and the negative control group, employing Image-Pro Plus 6.0 software.

## Statistical analysis

We analyzed the experimental data from more than 3 independent experiments per group with the analysis of variance (ANOVA) module of SPSS16.0 statistical software. The data are expressed as the mean $\pm$ standard deviation (SD).

# RESULTS

## Relative expression and localization of JARID2 in bovine oocytes and early embryos

To compare the differences in expression between JARID2 bovine oocytes at the GV and MII stages and early embryos at the 2-cell, 4-cell, 8-cell, 16-cell and blastocyst stages, we designed primers for JARID2 and measured the JARID2 expression level with qRT-PCR. The experimental results showed that JARID2 was expressed in bovine oocytes at the GV
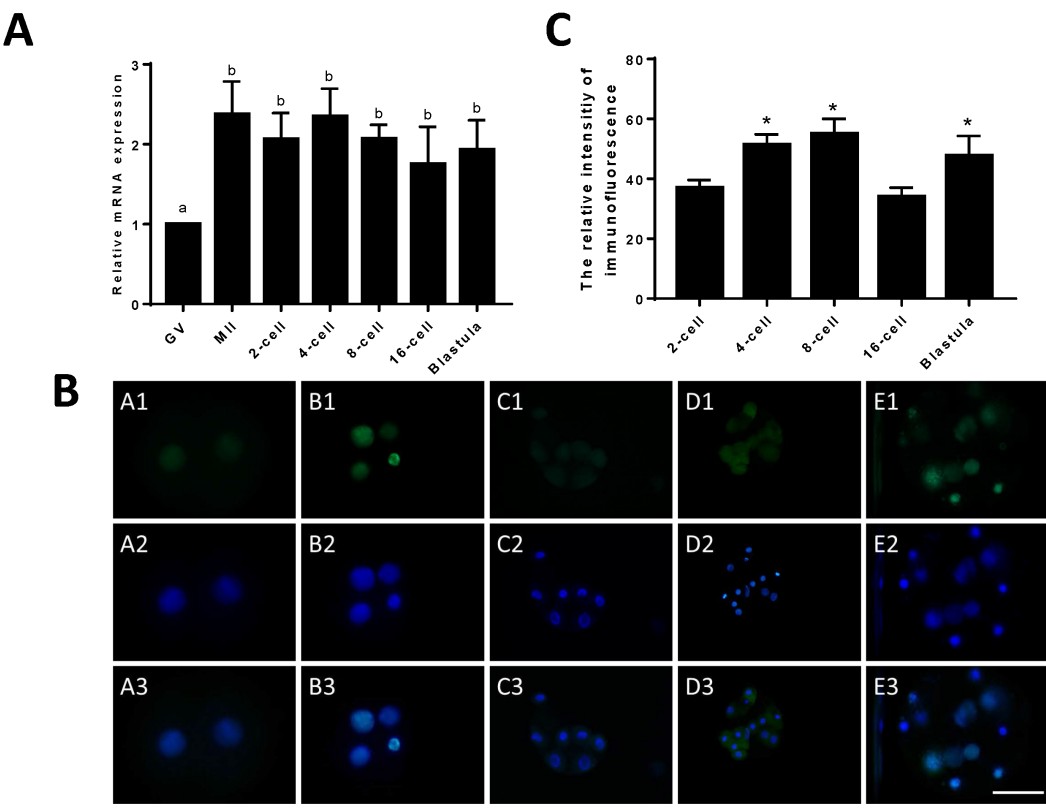

**Figure 1** **Relative expression of JARID2 in bovine oocytes and early embryos.** (A) Relative expression of JARID2 mRNA in oocytes at the GV stage and embryos at the 2-cell, 4-cell, 8-cell, 16-cell and blastula stages. JARID2 is expressed at all oocyte and early embryonic stages, and its expression at the MII stage and early embryonic stages is significantly higher than that at the GV stage ($p < 0.05$). (B) Immunofluorescence localization of JARID2 protein in early embryonic development. In A1 $\sim$ E1, green FITC fluorescence represents the location of JARID2 protein at the 2-cell, 4-cell, 8-cell, 16-cell and blastula stages, and A2 $\sim$ E2 blue DAPI fluorescence represents the location of DNA in the cell nucleus. A3 $\sim$ E3 are the merged images. (C) The immunofluorescence intensity of JARID2 protein in early embryonic development. The results show that JARID2 is expressed at all stages of early embryonic development. Data are presented as the mean $\pm$ SD derived from triplicate transfectants of three independent experiments. Scale bar = 100 $\mu$m. Different superscripts (a and b) denote a statistically significant difference. * denotes a statistically significant difference ($p < 0.05$).

and MII stages and in early embryos at the 2-cell, 4-cell, 8-cell, 16-cell and blastocyst stages. JARID2 expression was lowest in oocytes at the GV stage, where it was significantly lower than that in oocytes/embryos at other stages ($p < 0.05$). There was no significant difference in JARID2 expression among oocytes/embryos at other stages ($p > 0.05$) (Fig. 1A).

Immunofluorescent staining was performed on early bovine embryos at the 2-cell, 4-cell, 8-cell, 16-cell and blastocyst stages. We found that JARID2 was expressed at all stages of embryonic development and was mainly expressed in the cell nucleus (Fig. 1B).

We measured the fluorescence signals with a fluorescence microscope and collected images using the same parameters. In addition, we determined the mean fluorescence intensity at the embryonic nuclei to represent the amount of JARID2 expression at

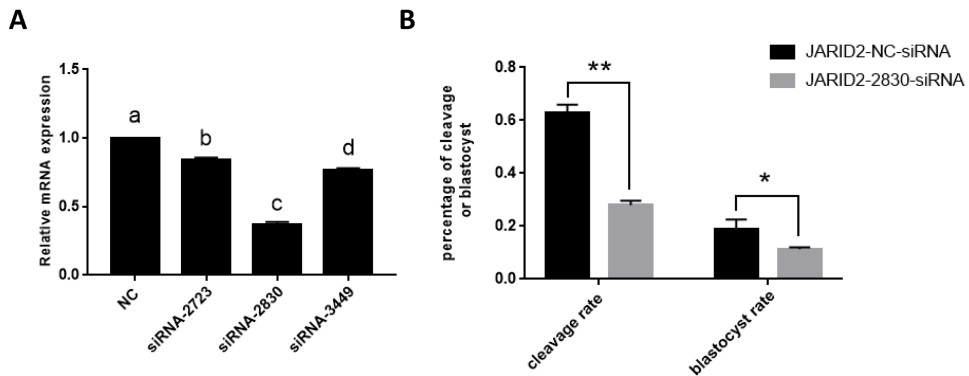

**Figure 2** **The influence of JARID2 siRNA interference on early embryonic development.** (A) The silencing effects of siRNA targeting different loci on JARID2 in bovine cumulus cell. Among the three siRNAs, siRNA-2830 has the highest silencing efficiency. After microinjection, the expression of JARID2 is significantly lower in the siRNA-2830-injected group by about 65% than the negative control group ($p < 0.05$); therefore, siRNA-2830 is suitable for subsequent RNA interference. (B) The effect of JARID2 silencing through microinjection of siRNA-2830 on early embryonic development. The cleavage rate are extremely significantly lower ($p < 0.01$) and blastocyst rate are significantly lower ($p < 0.05$) in the siRNA-treated group than in the negative control group. Data are presented as the mean ± SD derived from triplicate transfectants of three independent experiments. Different superscripts (a and b) denote a statistically significant difference. * denotes a statistically significant difference ($p < 0.05$), and ** denotes an extremely significant difference ($p < 0.01$).

different stages of early embryonic development. The mean fluorescence intensity of the embryonic nuclei was measured at the 2-cell, 4-cell, 8-cell, 16-cell and blastula stages with Image-Pro Plus 6.0. The results are shown in Fig. 1C. There was no significant difference in the expression of JARID2 between the 2-cell and 16-cell stages, but expression in the other stages of early bovine embryos was significantly higher than that in the 2-cell stage ($p < 0.05$). The expression of JARID2 determined based on the fluorescence intensity was consistent with the results of qRT-PCR analysis (Fig. 1C).

## Effect of JARID2 siRNA interference on early embryonic development

The relative expression of JARID2 mRNA in the bovine cumulus cells after siRNA silencing was detected via qRT-PCR, as shown in Fig. 2. Compared with the negative controls, siRNA-2830 had the greatest inhibitory effect on JARID2—approximately 65% ($p < 0.05$). Therefore, siRNA-2830 was used in subsequent experiments (Fig. 2A).

The expression of JARID2 was subsequently knocked down by microinjection of siRNA into the zygotes. After microinjection, we performed IVC, collected embryos at all stages, evaluated embryonic development and conducted statistical analyses of the cleavage rate and blastocyst rate. The statistical analyses showed that the cleavage rates were substantially decreased ($p < 0.01$), and blastocyst rates were significantly decreased ($p < 0.05$) (Fig. 2B).

## Effect of JARID2 on embryonic development-related gene expression

IVC of zygotes and collection of early embryos at the 2-cell stage and blastula stage, followed by detection of the mRNA expression levels of embryonic development-related genes (OCT4, SOX2 and c-myc), showed that these genes were expressed at higher levels

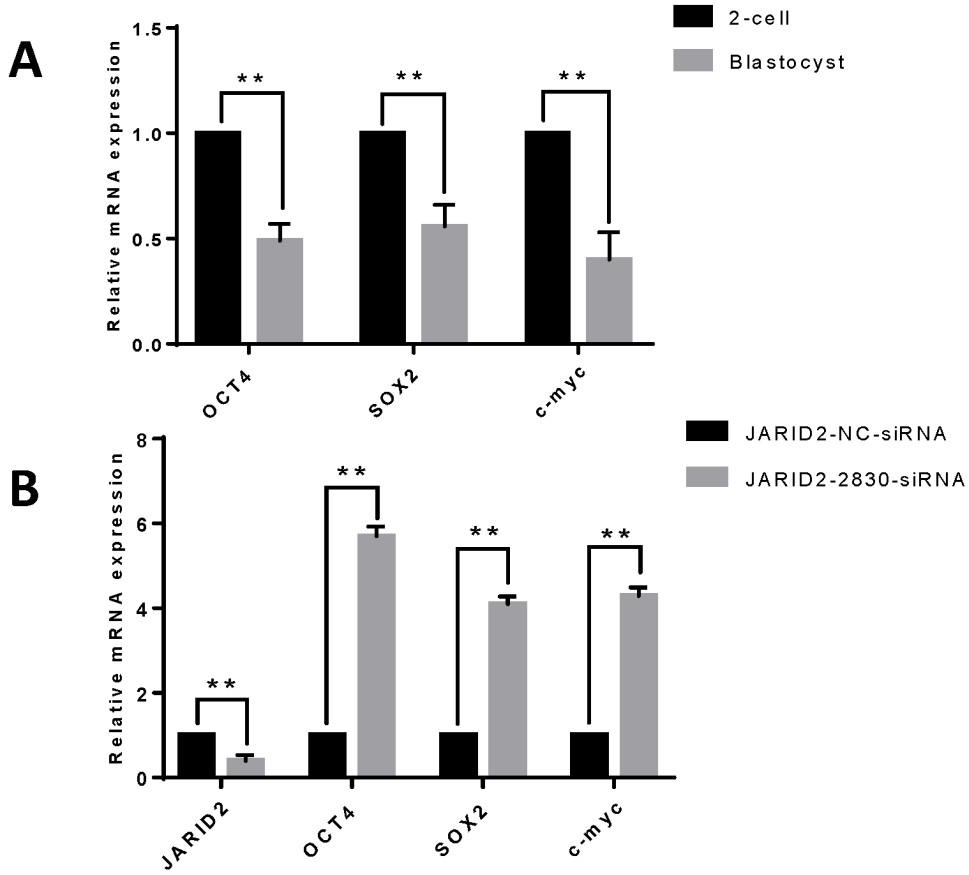

**Figure 3** **The effect of JARID2 on embryonic development-related gene expression.** (A) Relative mRNA expression of OCT4, SOX2 and c-myc in embryos at the 2-cell and blastocyst stages. At the blastocyst stage, expression of the three genes is extremely significantly lower than that at the 2-cell stage ($p < 0.01$). (B) Relative expression of embryonic development-related genes after transfection of siRNA-2830 to knockdown JARID2 expression. JARID2 expression is extremely significantly reduced ($p < 0.01$) and OCT4, SOX2 and c-myc expression is extremely significantly increased ($p < 0.01$) in the siRNA-2830-transfected group compared with the control group. Data are presented as the mean $\pm$ SD derived from triplicate transfectants of three independent experiments. ** denotes an extremely significant difference ($p < 0.01$).

in embryos at the 2-cell stage than in embryos at the blastula stage (Fig. 3A). Changes in embryonic development-related gene expression were also detected after JARID2 knockdown in early embryos at the 2-cell stage, which showed that the mRNA expression levels of OCT4, SOX2 and c-myc were highly significantly increased (Fig. 3B).

## JARID2 knockdown affects histone H3K27 methylation at the 2-cell, 4-cell, 8-cell and blastula stages

We examined the fluorescence intensity of H3K27me3, representing the expression level of H3K27me3, in the early embryos of the JARID2 siRNA silencing group and the negative control group following immunofluorescence staining. The results showed that the expression of H3K27me3 in the JARID2 siRNA silencing group at the 2-cell stage,

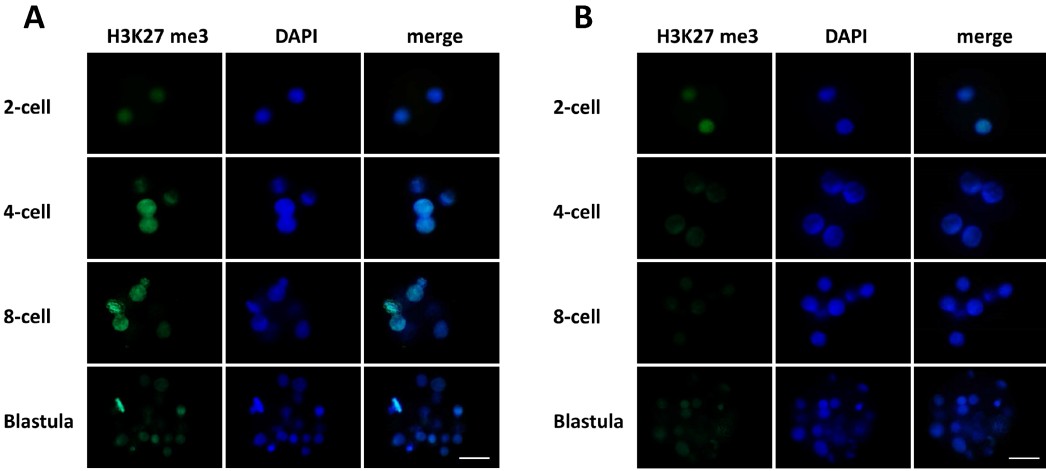

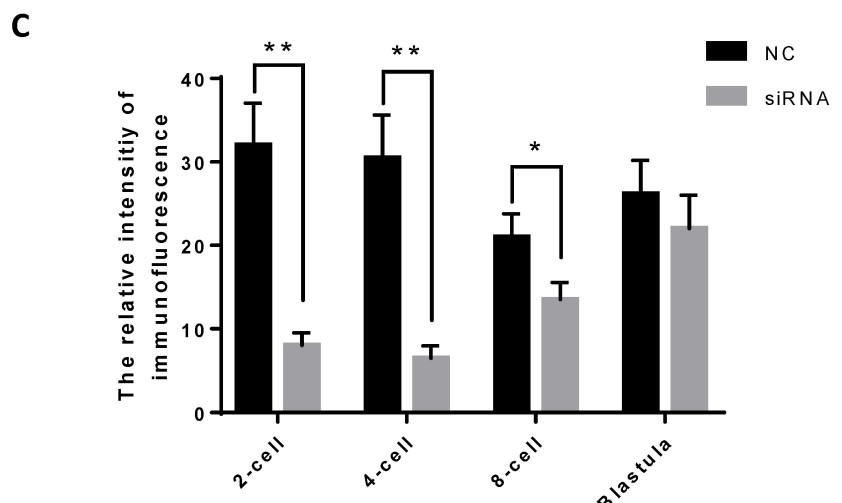

**Figure 4** **JARID2 knockdown affects histone H3K27 methylation at the 2-cell, 4-cell, 8-cell and blastula stages.** (A) H3K27me3 immunofluorescence staining in the negative control group; green FITC fluorescence represents H3K27me3 expression, and blue DAPI fluorescence represents the location of DNA in the cell nucleus. Scale bar $=100\,\mu$m. (B) H3K27me3 immunofluorescence staining in the JARID2 knockdown group. Scale bar $=100\,\mu$m. (C) Immunofluorescence intensity analysis. H3K27 methylation of the knockdown group at the 2-cell, 4-cell stages are extremely significantly lower than that of the negative control group at the same stages ($p < 0.01$), and at 8-cell stages are significantly lower ($p < 0.05$) than control group; however, at the blastula stage, H3K27 methylation is not significantly different between the two groups ($p > 0.05$). Data are presented as the mean $\pm$ SD derived from triplicate transfectants of three independent experiments. * denotes a statistically significant difference ($p < 0.05$), and ** denotes an extremely significant difference ($p < 0.01$).

4-cell stage and 8-cell stage was significantly lower than that in the control group at the same stages. Additionally, there was no change in H3K27me3 expression in the blastulas of the JARID2 siRNA silencing group compared with the blastulas of the negative control group (Figs. 4A–4B).

After microinjection of JARID2 2830 siRNA, the degree of H3K27me3 expression was significantly reduced compared with the controls, but there was no significant difference in H3K27me3 expression at the blastula stage (Fig. 4C).

## DISCUSSION

JARID2, as a member of the jumonji histone family, has a conserved DNA structural domain and could play an important regulatory role in normal early embryonic development as a regulator of several key developmental genes. JARID2 and other members of the jumonji histone family exhibit only 30% similarity in their spatial structures; therefore, their main functions may be markedly different from other members of the family (*Gregory et al., 1996*; *Iwahara & Clubb, 1999*; *Takeuchi, 1997*). Within the histone demethylase family, JARID2 is a special histone that does not show demethylase activity but can elicit histone methyltransferase activity by specific means. JARID2 mainly wraps around EZH2, a member of the PRC2 family, and exerts its methyltransferase activity to affect the maturation rate of oocytes and the cleavage rate of early embryos (*Lee & Skalnik, 2005*; *Zofall & Grewal, 2006*). Patients with a chronic malignancy are deficient in the JARID2 protein, which may result in the loss of enzymatic activity during early embryonic development and impede the normal development of embryos (*Li et al., 2010*; *Puda et al., 2012*). In this study, through qRT-PCR, immunofluorescence staining and fluorescence intensity analyses, the expression level and pattern of JARID2 were evaluated in bovine oocytes at the GV and MII stages and in bovine embryos at the 2-cell, 4-cell, 8-cell, 16-cell and blastocyst stages. JARID2 was highly expressed in oocytes at the GV and MII stages and in embryos at the 2-cell, 4-cell, 8-cell, 16-cell morula and blastula stages.

JARID2 expression was lower in oocytes at the GV stage than in the other examined stages, and the difference was statistically significant. The above results show that with the development of oocytes and embryos, the levels of JARID2 gene expression increased. These findings indicates that JARID2 may play an important role in the development of early bovine embryos and that embryos develop correctly only in the presence of high levels of JARID2, demonstrating the value of this research.

It has been reported that JARID2 plays an important role in the differentiation of early ESCs as a recruiter of PRC2, which regulates transcriptional translation primarily through the fine adjustment of chromatin structure to facilitate normal embryonic development (*Landeira et al., 2010*). JARID2 target sites are rich in CGG and GA sequences (*Peng et al., 2009*), and the DNA winding sequence irregularly appears in chromatin (*Li et al., 2010*), which is consistent with the finding of the present study that JARID2 is mainly expressed in the cell nucleus. JARID2 expression levels represented by the fluorescence intensity were consistent with the results of real-time fluorescent quantitative PCR. After confirming that JARID2 is a nuclear protein, we began to consider what role JARID2 plays in the cell nucleus. JARID2 is important for the development of mouse embryos (*Kitajima et al., 1999*). Mice lacking the JARID2 protein exhibit developmental deficits, including abnormal development due to excessively fast cell differentiation and abnormal development in

cardiomyocytes and megakaryocytes (*Jung et al., 2005a*; *Lee et al., 2000*; *Mysliwiec, Bresnick & Lee, 2011*; *Takeuchi et al., 1999*; *Takeuchi et al., 1995*). In several studies, it was found that overexpression of the JARID2 protein can lead to significantly reduced cell proliferation and DNA synthesis (*Toyoda, Kojima & Takeuchi, 2000*). Our results also demonstrate that JARID2 acts as an important transcription factor or regulatory protein that affects the development of oocytes and embryos, and it may be involved in epigenetic modification.

The results of the present study showed that JARID2-siRNA-2830 could silence JARID2 expression with an interference efficiency of approximately 65% ($p < 0.05$), based on real-time fluorescent quantitative PCR. Therefore, JARID2-siRNA-2830 could be used in the subsequent experiments. After the knockdown of JARID2 through microinjection of JARID2-2830-siRNA, the cleavage rate and blastocyst rate of early bovine embryos in the JARID2-2830-siRNA silencing group were significantly decreased compared with the negative control group. These results suggest that JARID2 plays an important regulatory role in early bovine embryonic development and that a lack of JARID2 could inhibit the development of early embryos. Several studies have shown that JARID2 is closely related to the production of blastulas during early embryonic development and participates in the formation of organs (*Corcoran et al., 2011*; *Peng et al., 2009*; *Ulitsky et al., 2011*). During germ cell development, JARID2 interacts with most inhibitory complex 2 to regulate the self-renewal ability of ESCs, and the absence of JARID2 reduces the transferability of mouse hematopoietic stem cells and embryonic stem cells (*Cai et al., 2013*; *Margueron et al., 2009*; *Zhao et al., 2010*). Our results are consistent with previous studies. The JARID2 protein plays an indispensable role in early embryonic development. The absence of JARID2 hinders normal embryonic development, reducing not only the cleavage rate but also the blastocyst rate. These results suggest that JARID2 promotes early embryonic development. Embryos lacking JARID2 do not develop normally. Based on our research, we can speculate that JARID2 regulates embryonic development by regulating certain factors. Its functions and mechanisms shall be further explored and studied. To further study the mechanism of JARID2 in embryonic development, we examined the changes in embryonic development-related genes and H3K9 methylation after siRNA knockdown of JARID2.

Several papers have reported that the OCT4 gene is mainly expressed in ESCs and reproductive stem cells and plays an important role in maintaining the pluripotency and self-renewal ability of embryonic stem cells (*Looijenga et al., 2003*). The OCT4 gene expression level decreases with the differentiation of tissue cells (*Hough et al., 2006*). SOX2, c-myc and OCT4 have synergistic effects and are important transcription factors that can regulate the pluripotency and self-renewal ability of ESCs; moreover, they play an important role in early embryonic development (*Oster et al., 2002*). The expression level of SOX2 in early embryos at the 2-cell stage is higher than in early embryos at the blastula stage (*Rodda et al., 2005*). The expression level of c-myc is decreased after the 8-cell stage, which may be due to the different roles of SOX2 and c-myc in embryonic development. In the present study, JARID2 expression was silenced in zygotes obtained through *in vitro* fertilization via microinjection of JARID2-2830-siRNA. By detecting the mRNA expression levels of OCT4, SOX2 and c-myc at the 2-cell stage with qRT-PCR, we found that the

mRNA expression levels of OCT4, SOX2 and c-myc were highly significantly increased in the JARID2 knockdown group compared with the control group. Therefore, we speculate that JARID2 plays an important role in early embryonic development by regulating the expression of OCT4, SOX2 and c-myc.

The members of the jumonji demethylase family exhibit a JmjC structural domain, and most members can demethylate histone lysine using Fe(II) and α-ketoglutarate (αKG) as co-enzyme factors. As previously reported, JHDMs can demethylate H3K36 (JHDM1) and H3K9me1; JHDM1 and JHDM2A can demethylate H3K9 and H3K36; and JARID1 can demethylate H3K9me3 and H3K27me3. However, whether JARID2, a member of the jumonji protein family, shows catalytic functions or demethylates H3K27me3 is uncertain. Histone H3K27 methylation is an important modification in embryonic development, stem cell formation and tumor transition (*Margueron & Reinberg, 2011*). PRC2 complexes regulate H3K27 methylation to silence genes (*Chase & Cross, 2011*), and JARID2 promotes the recruitment of PRC2 to DNA via binding with PRC2 (*Li et al., 2010*; *Pasini et al., 2010*; *Peng et al., 2009*; *Shen et al., 2009*).

In this study, early embryos were collected at the 2-cell, 4-cell, 8-cell and blastocyst stages after *in vitro* maturation, *in vitro* fertilization and microinjection of JARID2-2830-siRNA. Immunofluorescence staining and fluorescence intensity analysis were performed to detect the expression of H3K27me3 after JARID2 interference, and the difference in expression was taken as a measurement index of changes in H3K27me3 levels. The experimental results showed that the expression levels of H3K27me3 at the 2-cell stage, 4-cell stage and 8-cell stage in the JARID2 knockdown group were highly significantly decreased, whereas the level of H3K27me3 at the blastula stage was not significantly changed. The lack of change in the H3K27me3 level at the blastula stage might be due to the timing of JARID2 siRNA microinjection; after silencing JARID2 at the zygote stage, the silencing efficiency may gradually decrease with embryonic development. It was previously reported that JARID2 interacts with histone methyltransferases at the molecular level or regulates the expression of key genes in embryonic development, thus promoting early embryonic development (*Peng et al., 2009*; *Rinn et al., 2007*; *Rostovskaya et al., 2012*; *Son et al., 2013*; *Yuan et al., 2012*). During the process of oocyte meiosis, H3K27me3 may be dynamically changed, which could impact early embryonic development (*Kaneko et al., 2014*; *Kim et al., 2004*; *Landeira & Fisher, 2011*).

## CONCLUSION

This study confirmed that JARID2 is expressed in the cell nucleus of bovine oocytes and early embryos *in vitro* and that JARID2 interference inhibits embryonic development, reduces the cleavage rate and blastocyst rate and reduces the methylation level of H3K27. In addition, JARID2 could affect early embryonic development by modulating OCT4, SOX and c-myc expression. Our study results may provide a theoretical basis for discussing the main functions of the interaction between histone lysine methylation and key genes involved in early embryonic development in bovine oocytes and early embryos.

### Funding

This study was supported by the National Natural Science Foundation of China (31572400, 31501954) and the earmarked fund for China Agriculture Research System (CARS-37). The funders had no role in study design, data collection and analysis, decision to publish, or preparation of the manuscript.

### Grant Disclosures

The following grant information was disclosed by the authors:
National Natural Science Foundation of China: 31572400, 31501954.
China Agriculture Research System: CARS-37.

### Competing Interests

The authors declare there are no competing interests.

### Author Contributions

- Yao Fu performed the experiments, analyzed the data, wrote the paper, prepared figures and/or tables, reviewed drafts of the paper.
- Jia-Jun Xu, Xu-Lei Sun and Chang Liu performed the experiments, reviewed drafts of the paper.
- Hao Jiang and Yan Gao analyzed the data, contributed reagents/materials/analysis tools, reviewed drafts of the paper.
- Dong-Xu Han performed the experiments, wrote the paper, prepared figures and/or tables, reviewed drafts of the paper.
- Bao Yuan and Jia-Bao Zhang conceived and designed the experiments, wrote the paper, reviewed drafts of the paper.

### Animal Ethics

The following information was supplied relating to ethical approvals (i.e., approving body and any reference numbers):

The experiments were strictly performed according to the guidelines of the Guide for the Care and Use of Laboratory Animals of Jilin University. In addition, all experimental protocols were approved by the Institutional Animal Care and Use Committee of Jilin University (Permit Number: 20160522).

### Data Availability

The raw data has been supplied as Data S1.

### Supplemental Information

Supplemental information for this article can be found online at http://dx.doi.org/10.7717/peerj.4189#supplemental-information.

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
