# Peer review of "Function of JARID2 in bovines during early embryonic development"

_PeerJ, doi:10.7717/peerj.4189_

## Round 0.1 · original submission · Minor Revisions

Please revise the manuscript according to the reviewers' comments.

Reviewer 1 ·

Basic reporting

Please double-check the grammatical mistakes in this paper.

Experimental design

Design good

Validity of the findings

The paper is concise, well written and contains some interesting findings.

Additional comments

In this paper, authors investigated the function of JARID2 during bovine embryonic development. They studied the expression pattern during early embryonic process by using qRT-PCR and immunofluorescence and found that the cleavage rate and blastocyst rate were decreased after small interference mediated knocking down of JARID2, which is maily caused by dysregulation of pluripotent related genes expression in 2-cell stage. Finally, they detected the H3K27me3 level and found its level was significantly decreased in all the early embryonic stage.


1. The authors used double-stranded RNA to knockdown JARID1, did the authors detect the change of mRNA level of JARID1 after knockdown?
2. Manuscript should be written clearly in uniform line spacing.
3. Please move “how to analyse the fluorescence intensity” in line 257-258 into the M&M section.
4. Please double-check the grammatical mistakes in this paper.

Reviewer 2 ·

Basic reporting

This study identified the expression pattern of JARID2 in bovine oocytes and early embryonic stages and suggested that JARID2 played a key role in early embryonic development by regulating the expression of OCT4, SOX2 and c-myc via the modification of H3K27me3 expression. These results provide new data for the improvement in bovine in vitro embryo culture and understanding the regulation mechanisms in early embryonic development. However some contents need more modifications and explanations.

1. Some common mistakes or different words in English writing. These should be modified. I suggest that authors should check the manuscript carefully and try to avoid any mistakes like these:
Line 88 “MII stage”, but “M2 stage” in Fig 1 legend.
Line 181, “antibody (Abcom)”, is it “Abcam”? Company name and country should add to the manuscript like Line 161 “siRNA (Gene Pharma, China)”.

2. The Introduction Part.
The authors do not have to write the Introduction contents with extra title like “1. DNA methylation” and “2. Histone modification”. I suggest the authors rewrite this small part based on the important factors of histone modification.
3. Fig 2B. The description of Y-axis is missing.
In Fig 1B, Fig 4A and Fig 4B, the magnification or standard bar should be provided.

4. The Discussion Part.
This part should be more improved. The authors use lots of contents to describe the function of JARID2, H3K9 and H3K27. I think these contents should belong to the Introduction. More discussion like what’s its potential meaning of expression in cell nucleus and the synergy/different effects from this study would attract more readers’ attention. I think contents in line 314-329 are useful.

Experimental design

Experimental design is well done.

Validity of the findings

No comment

Reviewer 3 ·

Basic reporting

This manuscript investigated the function of JARID2 during bovine early embryonic development. JARID2 has been shown to be an important epigenetic regulator for embryonic development via histone modification. Although there have been many reports on the function of JARID2 during early embryonic development, the current study systemically examined the expression pattern of JARID2 in bovine and analyzed its possible regulatory roles on the expression of genes important to the maintenance of stem cell.
1. English writing
In general, the manuscript is clear, intelligible and professional. However, the writing could be greatly improved if the manuscript could be reviewed by a native English speaker. The followings are some examples or suggestions:
A. It would be consistent if the authors could use past tense to present or discuss their data and use present tense to discuss published data. However, it seems that the tense is randomly used, not consistent.
B. Line 41: after knockdown JARID2…………..after knockdown of JARID2 or after JARID2 knockdown.
C. Line 42: each early embryonic stages, suggest that JARID2 plays a key role in early embryonic development…………………early embryonic stages, suggesting that
D. Line 55: Early embryonic development is when cell division and differentiation are most active and when there is large-scale transcription of zygotic genes……….. Early embryonic development is characterized by the most active cell division and differentiation with large-scale transcription of zygotic genes.
E. Line 261: the jumonji histone family….. the jumonji histone demethylase family
F. Line 266: JARID2 is a special histone…………. JARID2 is a special histone demethylase member
2. Introduction
The introduction is well referenced. However, the introduction could be better structured by making the general introduction of epigenetic regulation more concise and focusing on JARID2 related studies, especially in terms of what is not known. In addition, the significance of the current study should be better defined.
3. Discussion
The discussion could be more concise with focus on the significance of current findings.
4. Legend for figure 2-4. Present tense should be used in figure legend.

Experimental design

The experiment is well designed with valid controls.

Validity of the findings

The results are sound with valid statistical analysis.

The quantity of siRNA in cells of blastocysts should be much lower due to siRNA partition during cell division. It should be cautious in explaining the gene expression data without knowing whether enough siRNA exists to interfere with JARID2 in cells of blastocysts.

Additional comments

No comment

---

## Round 0.2 · accepted · Accept

I think all the concerns raised by the reviewers were addressed, especially the writing of Introduction and Discussion part were improved.